# A pilot randomized trial of incentive strategies to promote HIV retesting in rural Uganda

Gabriel Chamie[1]*, Alex Ndyabakira[2], Kara G. Marson[1], Devy M. Emperador[1], Moses R. Kamya[3], Diane V. Havlir[1], Dalsone Kwarisiima[4], Harsha Thirumurthy[5]

**1** University of California, San Francisco, California, United States of America, **2** Infectious Diseases Research Collaboration, Mbarara, Uganda, **3** Makerere University College of Health Sciences, Kampala, Uganda, **4** Makerere University Joint AIDS Program, Kampala, Uganda, **5** University of Pennsylvania, Philadelphia, Pennsylvania, United States of America

\* Gabriel.chamie@ucsf.edu

## Abstract

### Background

Retesting for HIV is critical to identifying newly-infected persons and reinforcing prevention efforts among at-risk adults. Incentives can increase one-time HIV testing, but their role in promoting retesting is unknown. We sought to test feasibility and acceptability of incentive strategies, including commitment contracts, to promote HIV retesting among at-risk adults in rural Uganda.

### Methods

At-risk HIV-negative adults were enrolled in a pilot trial assessing feasibility and acceptability of incentive strategies to promote HIV retesting three months after enrollment. Participants were randomized (1:1:3) to: 1) no incentive; 2) standard cash incentive (~US$4); and 3) commitment contract: participants could voluntarily make a low- or high-value deposit that would be returned with added interest (totaling ~US$4 including the deposit) upon retesting or lost if participants failed to retest. Contracts sought to promote retesting by leveraging loss aversion and addressing present bias via pre-commitment. Outcomes included acceptability of trial enrollment, contract feasibility (proportion of participants making deposits), and HIV retesting uptake.

### Results

Of 130 HIV-negative eligible adults, 123 (95%) enrolled and were randomized: 74 (60%) to commitment contracts, 25 (20%) to standard incentives, and 24 (20%) to no incentive. Of contract participants, 69 (93%) made deposits. Overall, 93 (76%) participants retested for HIV: uptake was highest in the standard incentive group (22/25 [88%]) and lowest in high-value contract (26/36 [72%]) and no incentive (17/24 [71%]) groups.

### Conclusion

In a randomized trial of strategies to promote HIV retesting among at-risk adults in Uganda, incentive strategies, including commitment contracts, were feasible and had high

**Data Availability Statement:** All relevant data are available at https://doi.org/10.7910/DVN/KYQPME

**Funding:** This work was supported by a grant (R01MH105254) from the National Institute of

Mental Health (NIMH) at the National Institutes of Health (GC, HT). The funders had no role in study design, data collection and analysis, decision to publish, or preparation of the manuscript.

**Competing interests:** The authors have declared that no competing interests exist.

acceptability. Our findings suggest use of incentives for HIV retesting merits further comparison in a larger trial.

## Trial registration

ClinicalTrials.gov identifier: NCT:02890459

## Introduction

As global efforts continue to reduce HIV incidence through combination prevention approaches (such as pre-exposure prophylaxis, voluntary medical male circumcision, condoms, and universal HIV treatment as prevention), it is clear that universal HIV testing initiatives must be followed by targeted testing services that offer frequent retesting for HIV to individuals who test HIV-negative but remain at risk. Retesting of high-risk, HIV-uninfected persons is critical to identifying those recently infected with HIV, to maximize individual benefits of early antiretroviral treatment and reduce onward transmission [1–3]. HIV retesting also offers an opportunity to reinforce behavioral and biomedical prevention among at-risk populations [4]. The World Health Organization recommends HIV retesting at least annually for all sexually active adults living in high HIV burden settings, including key populations at increased risk of HIV infection, noting that more frequent testing (i.e. every 3–6 months) may be warranted based on individual risks [4, 5]. In Uganda, where the HIV prevalence among 15-49-year-old adults is 6% [6], the Ministry of Health (MoH) recommends retesting of HIV-negative key populations every three months [7].

Despite guidelines recommending retesting for HIV among at-risk adults who previously tested HIV negative, published data across multiple settings in sub-Saharan Africa, including Uganda [8, 9], suggest that retesting annually (let alone more regular intervals) occurs infrequently, with most adults who access HIV testing reporting that they have not had an HIV test in the prior year [10–12]. In a qualitative study examining perceptions of HIV retesting prior to a universal "test and treat" intervention in South Africa, most participants expressed the view that retesting was unnecessary, particularly if a person continued to feel healthy following a prior negative HIV test [13]. In spite of the importance of retesting high-risk, HIV-negative adults at regular intervals to identify recent seroconversions and to reinforce prevention messaging, there are few evidence-based interventions to promote retesting among high-risk adults [14]. Optimal strategies to promote HIV retesting of high-risk adults in sub-Saharan Africa are unclear.

Economic incentives have been shown to promote a number of behaviors, including one-time HIV testing, in which the costs of a given behavior (e.g. stigmatization, transport costs and lost wages when accessing HIV testing) are immediate but the gains (e.g. the health benefits of antiretroviral therapy if HIV-infected or combination prevention if HIV-uninfected) may not be realized until later [15, 16]. As such, incentives may be effective in promoting retesting for HIV in people who test HIV antibody negative but remain at high-risk. However, in contrast to first-time HIV testing, interventions to promote retesting for HIV among high-risk individuals may face additional challenges, as motivation to retest may be relatively lower following a recent negative HIV test and benefits of accessing testing may be perceived as low, particularly if transportation to clinic and access to testing are costly [13]. Whether incentives can promote retesting for HIV in high-risk groups compared to counseling to retest alone (the current standard of care in many settings, such as Uganda [7]), and if so, what type of incentive approach is most effective for HIV retesting, remains unknown.

A common approach to implementing economic incentives for healthy behaviors has been to provide monetary or non-monetary rewards conditional on undertaking a desired behavior [16, 17]. However, recent studies have indicated that incentives can be made more effective by leveraging loss aversion, the tendency for losses to have a greater psychological impact on individuals than gains of comparable size [18, 19]. Thus, avoiding a monetary loss (such as a small financial penalty for not exercising) tends to be more motivating than a gain (such as a small financial gain for exercising) of equal amount [20, 21]. Individuals may also be aware of their own tendency towards present-biased decision-making: the tendency to place disproportionate weight on present (or short-term) rather than future (or long-term) costs and benefits [22]. One approach to putting loss aversion into practice, and to actively counter-acting present bias, is a "commitment contract" in which individuals interested in achieving a goal (such as smoking cessation, weight loss, or retesting for HIV) commit to the goal in advance by making a financial deposit, and then risk losing the deposit if they do not achieve the goal [18, 23, 24]. Efforts to implement commitment contracts in middle- and high-income countries have generally found that though there may be low uptake of commitment contracts, effectiveness can be high among those who agree to make a deposit [19, 23, 25]. However, to our knowledge, no incentive-based approaches in sub-Saharan Africa have attempted to leverage loss aversion for behavior change through commitment contracts, given legitimate concerns regarding acceptability and feasibility. The aim of this study was to assess the feasibility and acceptability of various incentive strategies, including commitment contracts that required a baseline deposit, to promote HIV retesting among at-risk adults in rural Uganda.

## Methods

We recruited adults at increased risk of HIV compared to the general population from a community in rural, southwestern Uganda (NCT:02890459). Study staff first met with local health officials and community representatives to identify venues frequented by adults considered at increased risk of HIV, due to attendance at bars associated with transactional sex, participation in sex work, high mobility to and from the community, or trading center work [7]. The venues identified included bars, motorcycle taxi stages, a hair salons, a restaurant, a primary care clinic waiting room, and a local trading center. Study staff then visited these venues to distribute recruitment cards inviting adults to come to the local government-run clinic for a free health evaluation that included HIV testing. Adults were asked to bring the recruitment cards in order to receive a one-time incentive payment of 15,000 Ugandan Shillings (USh; or ~US$4 in 2017) for the health evaluation and to consider joining a study. Adults who presented to the clinic with recruitment cards underwent rapid HIV antibody testing with the Alere Determine™ HIV-1/2 Ag/Ab Combo, followed by the Chembio HIV 1/2 STAT-PAK® and SD Bioline HIV-1/2 assays for confirmation of Determine-positive results according to Uganda MoH guidelines [7], as well as screening for hypertension and diabetes.

Enrollment was offered to eligible adults, 18–59 years of age, who presented with recruitment cards, tested negative for HIV, and had no intention of moving away from the community within three months. Eligible adults who provided informed, written consent underwent a baseline questionnaire: a 62-question survey developed by study investigators and administered by study staff to understand participant demographics, socioeconomic status, HIV risk behaviors, prior HIV testing behavior and testing preferences (see Supporting Information: S3). The questionnaire was followed by randomization (by block randomization, with block size = 10 and allocation sequence computer-generated prior to initiation of trial enrollment by study investigators). Participants chose a sealed envelope from several offered by study staff to unveil study group allocation to one of three groups (1:1:3) to promote HIV retesting within

2–3 months: a) no incentive (control); b) a standard, gain-framed (i.e. presented as a "gain" or bonus) cash incentive (15,000 USh); and c) a commitment contract, in which participants could voluntarily make a deposit that would be returned with added interest (totaling 15,000 USh) upon retesting. The contract group had low- and high-value deposit arms (2,500 USh = US$0.71, or 5,000 USh = US$1.42) in order to assess how the initial deposit amount affected uptake of the commitment contract. Participants in the contract groups were told the deposits were completely voluntary (i.e. there was no requirement to make a deposit) and that the deposits would be lost if they did not return for HIV retesting between 2–3 months following randomization. All groups received counseling encouraging retesting for HIV three months after enrollment, consistent with Uganda MoH guidelines for high-risk populations [7], at the same clinical site where baseline testing occurred. Outcomes included acceptability of trial participation (defined as the number of eligible adults that agreed to continue participating in the pilot trial following randomization) among eligible adults, feasibility of commitment contracts for HIV retesting (defined as the proportion of contract group participants who made a baseline deposit), HIV retesting uptake 2–3 months following enrollment (to determine likely effect sizes), and HIV seroconversion.

## Statistical analysis

Descriptive statistics were used to present baseline characteristics, including means, standard deviations (SD), medians, and interquartile ranges (IQR). Outcomes were calculated using exact methods to obtain proportions with binomial confidence intervals (CI). As this was a pilot study, we determined sample size based on an *a priori* interest in feasibility of commitment contracts in approximately 70 adults rather than using power calculations, in preparation for a larger trial of the effectiveness of incentive strategies to promote HIV retesting.

All participants provided written informed consent in their preferred language (English or Runyankole, the local language in the region). The Makerere University School of Medicine Research and Ethics Committee (Uganda), the Uganda National Council for Science and Technology, and the University of California San Francisco (UCSF) Committee on Human Research [26] approved the study protocol. The UCSF Committee on Human Research served as the institutional review board of record for investigators from the University of Pennsylvania [26].

## Results

From August 8–22 2017, study staff distributed 164 recruitment cards at 26 venues frequented by adults considered high-risk for HIV: bars, sites of commercial sex work (i.e. a hair salon and a restaurant), a trading centers with mobile vendors, a primary care clinic waiting room, and transportation hubs (i.e. truck stops and motorcycle taxi stages). Over eight days following recruitment card distribution, 153 (93%) adults presented with recruitment cards and tested for HIV at the local government-run health center. Of those tested, median age was 31 years (IQR: 26–37), 113 (74%) were men, and 23 (15%) tested HIV-positive. HIV positivity varied by recruitment site: e.g. 0% (0/11) at the clinic waiting room, 11% (11/99) at transport hubs, 13% (4/21) at trading centers, 54% (7/13) at bars/sites of commercial sex work, and 11% (1/9) at storefronts associated with informal sex work.

Of 130 HIV-uninfected, eligible adults, 123 (95%) enrolled in the trial and were randomized (Fig 1): 74 (60%) to commitment contracts (38 and 36 in low- and high-value deposits, respectively), 25 (20%) to gain-framed incentives, and 24 (20%) to no incentive. Among commitment contract participants, 69/74 (93%; 95% confidence interval [CI]: 85–98%) adults provided an initial deposit: 36/38 (95%) in low-value and 33/36 (92%) in high-value groups (Table 1). Of

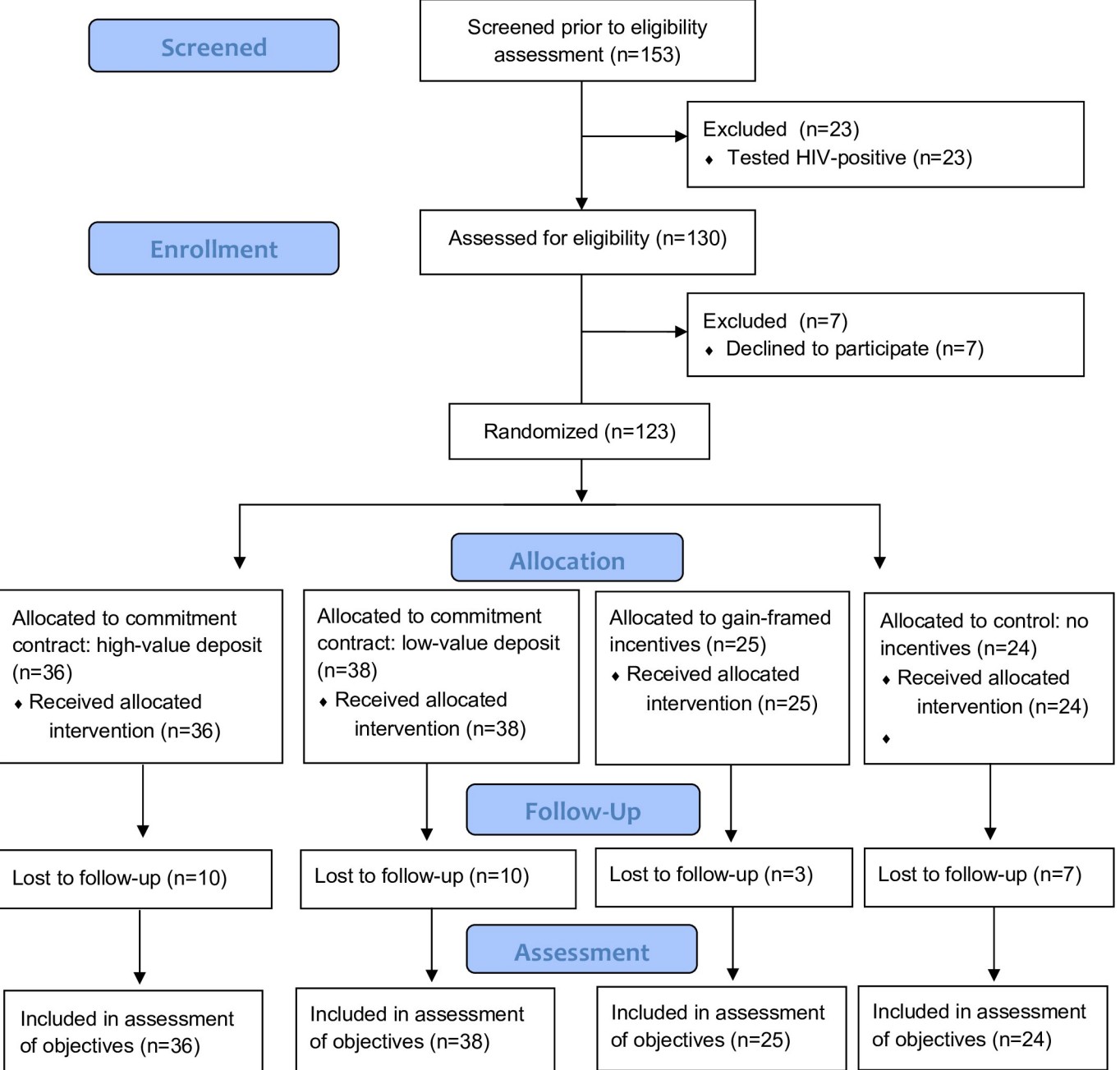

**Fig 1. Pilot trial CONSORT diagram, indicating the number of participants screened, enrolled, randomized with allocation to study arm, lost to follow up, and assessed at trial completion.**

the five participants who declined to deposit, reasons given included fear of losing the deposit (N = 2), and inability to afford the deposit amount (N = 3). Among the no incentive group and the gain-framed incentive group, 96% (23/24) and 92% (23/25) reported a willingness to provide a commitment contract deposit if offered, respectively.

Overall, 93/123 (76%; 95% CI: 67–83%) adults retested for HIV 2–3 months following their initial HIV-negative test. Retesting uptake was highest in the standard, gain-framed incentive group (88%) and lowest in high-value contract (72%) and no incentive (71%) groups (Table 1).

**Table 1. Baseline characteristics and HIV retesting uptake among at-risk adults (N = 123) enrolled in a pilot, randomized-controlled trial of incentives strategies to promote retesting for HIV three months after an initial negative HIV antibody test.**

| | No Incentive (Control) | Gain-framed Incentive | Commitment Contract: $0.71 deposit | Commitment Contract: $1.42 deposit |
|---|---|---|---|---|
| | (N = 24) | (N = 25) | (N = 38) | (N = 36) |
| **Age**: median (IQR) | 30 (28–36) | 30 (25–37) | 31 (25–37) | 31 (25–35) |
| **Female**: N (%) | 5 (21%) | 8 (32%) | 7 (18%) | 12 (33%) |
| **Daily wage (2017 US Dollars)**: mean (standard deviation (SD)) | $3.80 (3.64) | $2.59 (1.67) | $2.64 (1.92) | $2.45 (1.95) |
| **Attended Secondary school**: N (%) | 9 (39) | 9 (38) | 4 (11) | 12 (36) |
| **Married**: N (%) | 18 (75) | 18 (72) | 28 (74) | 27 (75) |
| **Tested for HIV ≤ one time in past 12 months, by self-report**: N (%) | 10 (42%) | 8 (32%) | 13 (34%) | 7 (19%) |
| **Self-reported having sex partner outside of primary relationship in past 12 months**: N (%) | 7 (29%) | 13 (52%) | 21 (55%) | 19 (53%) |
| **Self-reported number sex partners in past 12 months**: mean (SD) | 4.6 (2.8) | 2.8 (1.7) | 8.5 (20.8) | 2.9 (4.6) |
| **Received or paid money in exchange for sex in past 12 months, by self-report**: N (%) | 4 (17%) | 10 (40%) | 12 (32%) | 6 (17%) |
| **No condom use when receiving or paying money in exchange for sex in past 12 months, by self-report** | 2/4 | 4/10 | 4/12 | 5/6 |
| **Deposit made to commitment contract**: N (%; [95% CI]) | N/A | N/A | 36 (95% [82–99%]) | 33 (92% [78–98%]) |
| **Reported willingness to deposit, if offered opportunity**: N (%; [95% CI]) | 23 (96% [79–100%]) | 23 (92% [74–99%]) | N/A | N/A |
| **HIV Retesting 2–3 months following randomization**: N (%; [95% CI]) | 17 (71% [49–87%]) | 22 (88% [69–97%]) | 28 (74% [57–87%]) | 26 (72% [55–86%]) |

IQR = Interquartile Range; N/A = Not applicable; CI = Confidence Interval.

Among contract participants who made a deposit, HIV retesting was 78% in low- and 76% in high-value groups. As noted in the Methods, this pilot trial was not powered to detect statistically significant differences in retesting between arms. No seroconversions were observed.

## Discussion

In a pilot randomized controlled trial of financial incentives to promote retesting for HIV among at-risk adults in rural Uganda, incentive strategies, including commitment contracts with voluntary deposits, were feasible and had high (95%) acceptability. In this first attempt to implement commitment contracts to promote an HIV prevention behavior in sub-Saharan Africa, the large majority of participants (>90%) made a deposit committing to retesting in the future when offered the opportunity. Our findings suggest the use of incentives, including commitment contracts, for HIV retesting is feasible and merits further study in a larger trial.

Commitment contracts are an innovative and potentially low-cost approach to incentivizing behavior change that leverage loss aversion, the tendency of individuals to be more motivated by a loss than a gain of equal value when making decisions, as well as voluntary precommitment to overcome present bias [18]. While voluntary commitment contracts have been put into practice for behaviors such as smoking cessation or weight loss in high- and middle-income countries [19, 24, 25], to our knowledge, they have not been attempted in low-income countries where individuals' ability to make deposits may be limited by poverty. We sought to overcome these obstacles in a pragmatic manner by offering all participants a standard incentive for their initial HIV test, and then providing contract participants an opportunity to voluntarily deposit part of this initial incentive (or "endowment") if they wished to

commit to HIV retesting in the future. This approach of offering an opportunity to deposit part of this initial incentive also addressed a limitation observed in commitment contract programs in high- and mid-income countries, where a relatively small percentage of those offered such contracts actually make a deposit [19, 23]. The high rates of deposits made by participants in our study are encouraging in this regard and may indicate a clear understanding of the benefits of HIV retesting among high-risk individuals. Indeed, commitment contracts may be ideal for use by people who desire a long-term goal but need an additional "nudge" to achieve the goal. Alternatively, the high deposit rates observed could indicate that the deposit amount was too low–i.e. the loss of the deposit was a risk worth taking even if the desire to retest in the future was not strong. Though this latter explanation is possible, the deposit amounts represented between 25–50% of the average daily wage in the contract groups, suggesting the deposits were not trivial. Future studies of commitment contracts for HIV retesting may need to determine the optimal balance between deposit amounts that are acceptable and of sufficient value to generate loss aversion.

Our findings add to a growing literature on the use of strategies informed by behavioral economics, including financial incentives and opt-out provider-initiated testing and counseling, to promote HIV testing and other HIV-related behaviors and outcomes. The effectiveness of these strategies when studied in randomized trials has varied [27]. Examples of effective strategies have included gain-framed incentives to promote voluntary medical male circumcision (VMMC) in Kenya [28] and self-testing among male partners of women attending antenatal care in Malawi [29], as well as lotteries to promote HIV testing of adolescents in Zimbabwe [30], HIV testing among men in Uganda [9], and negative sexually transmitted infection screening results in Lesotho [31]. However, some studies have found financial incentives to be ineffective compared to no incentive control conditions for other HIV-related outcomes, such as achieving or maintaining viral suppression in Uganda [32] or decreasing engagement in high-risk sexual activity among men in Malawi [33]. The findings from this pilot trial add to this literature, by demonstrating that deposit contracts can be acceptable and feasible for promoting HIV retesting and might also be feasible for other health behaviors. Future research could address knowledge gaps in effectiveness of commitment contracts or incentives for retesting and explore barriers and facilitators to retesting among high-risk adults after a prior negative HIV test.

Our study has several limitations. First, our pilot trial was designed to test acceptability and feasibility of various incentive approaches, particularly commitment contracts, for HIV retesting and as such, was not powered to detect differences in retesting uptake by study arm or seroconversions. Second, although retesting uptake was high (>70%) in all groups, including the control group (71%), participants in the control group may have anticipated a reward upon retesting due to the receipt of the initial incentive at enrollment, despite our efforts to explain the absence of incentives for retesting. In future studies with more than one retesting opportunity, we suspect HIV retesting might decline over time when incentives are not offered, as has been observed with other repeated health behaviors such as smoking cessation [19]. In addition, whether participants who agree to commitment contracts would agree to a deposit without an initial endowment remains unknown. Third, our commitment contract group included "interest" on the initial deposit, and as such combined loss aversion with an additional gain. The interest was included to avoid a penalty for making deposits but may also explain the high acceptability of commitment contracts in this pilot. Fourth, no HIV seroconversions were observed in this study, possibly suggesting that we did not reach a high-risk population that needed frequent (i.e. every 3-month vs. annual) HIV retesting. However, participants reported high-risk sexual behavior including sex outside of a primary partnership and paying/receiving money in exchange for sex, in many cases without condom use. These

observations, along with the high prevalence of HIV among adults screened (15%, in a setting where the adult HIV prevalence is 6.5% among the general population [8]), suggest our recruitment strategy selected for high-risk individuals.

## Conclusion

As reported rates of "ever" having tested for HIV increase in sub-Saharan Africa and the number of first-time testers declines [34], identifying interventions to promote retesting in persons at increased risk for HIV becomes a greater priority. Our pilot data suggest that economic incentives, including commitment contracts, are an acceptable and feasible method to promote HIV retesting that merit further evaluation.

## Supporting information

**S1 Data. Deidentified study dataset.**
(DOC)

**S2 Data. Version of the study protocol that was in use at the time of initial participant enrollment in this study.**
(DOCX)

**S1 File.**
(PDF)

## Acknowledgments

We thank the residents of the study communities for their generous participation in our study. We gratefully acknowledge the contributions of Jane Kabami and Stella Kabageni from the Infectious Diseases Research Collaboration in Mbarara, Uganda.

## Author Contributions

**Conceptualization:** Gabriel Chamie, Dalsone Kwarisiima, Harsha Thirumurthy.

**Data curation:** Gabriel Chamie, Kara G. Marson, Devy M. Emperador.

**Formal analysis:** Gabriel Chamie, Kara G. Marson, Harsha Thirumurthy.

**Funding acquisition:** Gabriel Chamie, Moses R. Kamya, Harsha Thirumurthy.

**Investigation:** Gabriel Chamie, Moses R. Kamya, Diane V. Havlir, Dalsone Kwarisiima, Harsha Thirumurthy.

**Methodology:** Harsha Thirumurthy.

**Project administration:** Alex Ndyabakira, Kara G. Marson, Devy M. Emperador, Dalsone Kwarisiima.

**Supervision:** Gabriel Chamie, Moses R. Kamya, Dalsone Kwarisiima.

**Writing – original draft:** Gabriel Chamie.

**Writing – review & editing:** Gabriel Chamie, Alex Ndyabakira, Kara G. Marson, Devy M. Emperador, Moses R. Kamya, Diane V. Havlir, Dalsone Kwarisiima, Harsha Thirumurthy.

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
