## [Decision Letter · Decision Letter 0]

16 Dec 2019

PONE-D-19-27665

A pilot randomized trial of incentive strategies to promote HIV retesting

PLOS ONE

Dear Dr. Chamie,

Thank you for submitting your manuscript to PLOS ONE. After careful consideration, we feel that it has merit but does not fully meet PLOS ONE’s publication criteria as it currently stands. Therefore, we invite you to submit a revised version of the manuscript that addresses the points raised during the review process.

We would appreciate receiving your revised manuscript by 31 January 2020. To enhance the reproducibility of your results, we recommend that if applicable you deposit your laboratory protocols in protocols.io, where a protocol can be assigned its own identifier (DOI) such that it can be cited independently in the future. For instructions see: http://journals.plos.org/plosone/s/submission-guidelines#loc-laboratory-protocols

We look forward to receiving your revised manuscript.

Kind regards,

Giuseppe Vittorio De Socio, MD, PhD

Academic Editor

PLOS ONE

Journal Requirements:

2. Please provide additional details regarding participant consent. In the ethics statement in the Methods and online submission information, please ensure that you have specified whether consent was suitably informed.

Reviewers' comments:

Reviewer's Responses to Questions

**Comments to the Author**

1. Is the manuscript technically sound, and do the data support the conclusions?

Reviewer #1: Partly

Reviewer #2: Partly

2. Has the statistical analysis been performed appropriately and rigorously? 

Reviewer #1: Yes

Reviewer #2: Yes

3. Have the authors made all data underlying the findings in their manuscript fully available?

Reviewer #1: Yes

Reviewer #2: Yes

4. Is the manuscript presented in an intelligible fashion and written in standard English?

Reviewer #1: Yes

Reviewer #2: Yes

5. Review Comments to the Author

Reviewer #1: 1) Interesting study to incentivize retesting of negative individuals vs. incentivizing staying HIV negative. Considering the WHO 2015 Consolidated guidelines on HIV testing services and the upcoming update being released in December 2019 recommends that for most people who test HIV negative additional retesting to rule out being in the window period is not necessary (after 3 months). In addition, the WHO guidelines recommends retesting of individuals with ongoing high risk of HIV infection, not all key populations require retesting every 3 months. It would be helpful for the authors to address the above in the introduction and discussion.

2) The methods of offering incentives is good; however, please address why incentivizing retesting of negative individuals vs. behavior change to remain negative.

3) Also address why incentivizing retesting of HIV negative individuals in a resource limited setting that is focusing more on targeted HIV testing services (e.g. partner notification services, risk screening tools in outpatient settings) with linkages to treatment and/or prevention services.

4) Please address how incentivizing retesting of the study participants is better than current practice in clinics or other HIV testing locations that offer HIV testing services? Please include references that show that there is a need to incentivize retesting of individuals who are HIV negative. There is a lot of peer reviewed literature that shows most HIV negative individuals (with or without ongoing risk) do come back for retesting.

Reviewer #2: General Comments:

Background: The study authors outline how the use of financial incentives in the role of HIV re-testing have yet to be studied. While this might be the case, I think the authors could make a stronger case for why the use of financial incentives for retesting would expect to be any different than the use of financial incentives for HIV testing in general? For example, do the authors expect a different outcome than what research has shown with financial incentives for HIV testing? Please clarify.

Background: This manuscript would benefit from some additional references, particularly in the Background section to provide a stronger case for how the use of financial incentives for HIV retesting would be beneficial for the field. For example, what is the prevalence and/or incidence of HIV infection in Uganda? How often do people get retested in Uganda? If the study authors provided some brief background information about HIV and HIV retesting uptake in Uganda this might also increase clarity for the international reader as to why innovative interventions are needed in this country/region.

Discussion: Please provide some detail as to how the study findings are comparable to other studies that use financial incentives in low-and-middle income countries in the HIV field.

Discussion: Please provide a brief statement how the findings from this study could inform future research in the field, and also, remaining knowledge gaps and areas for future research.

Specific Comments:

Abstract: In the Results section, the authors have listed number/percentage to display one set of findings but then only percentages for the remaining findings. Please use number/percentage to be consistent.

Introduction: “Economic incentives have been shown to promote a number of behaviours…” This paragraph is not entirely clear. For example, is ‘counseling alone’ the standard of care? Please rephrase this sentence: “the costs of a given behavior are immediate but the gains may not be realized until later…” (could the authors please provide an example?). Also, with this sentence: “insights from behavioural economic theory suggest that individuals display loss aversion…” Perhaps the authors provide an example to increase clarity of “loss aversion”?

Introduction: “A standard approach to implementing…” Please provide some references here to increase clarity.

Introduction: “Individuals may also be aware of a tendency towards present-biased…” Please modify this sentence to increase clarity.

Introduction: What is “Leverage loss aversion”? Please clarify.

Methods: How long did the authors give the participants to be re-tested? Did participants have to come back at exactly 3 months post study enrollment for re-testing? Please add some study details to increase clarity.

Methods: “considered at increased risk of HIV, due to drinking, sex work” What does “due to drinking” mean in this case? Please remove or modify to increase clarity.

Methods: What was the questionnaire that participants were asked to complete? Did the questionnaire only enquire about participant demographics or also measure ‘feasibility’ and ‘acceptability’? How many questions were in the questionnaire? Who developed the questionnaire? Who administered the questionnaire? Was it self-administered?

Methods: Please specifically define ‘feasibility’ and ‘acceptability’ and provide a reference, if possible. Also, please define how these outcomes are measured. In the Discussion section, it states, “were feasible and had high acceptability”, however, it is not clear how these outcomes were measured (and what could be defined as ‘high acceptability’).

Methods: Please define “standard, gain framed cash incentive” to increase clarity. Similarly, what is a “gain framed” incentive?

Methods: Where did participants need to go to receive retesting? At one clinic site? At several clinic sites?

Methods: What is standard of care in this case? i.e. “no incentive (control)”

Methods: “we determined sample size…in feasibility of commitment contracts in approximately 70 adults rather than using power calculations…” Please consider modifying this section to increase clarity. It is not clear how the study authors determined a sample size of 70 to measure feasibility. Further, was the sample size determined to only measure feasibility or also acceptability? At this point, the study authors could state the minimum number of participants needed in each condition to be powered to detect statistical significant differences between conditions. The study authors could also emphasise in the Results section that the study was not powered to detect statistical significance between conditions, and in the Discussion section, mention that this is an area for future research (would be powered in future trial).

Methods: Could the study authors please provide greater details as to the venues in which the participants were recruited from? Is there any indication of how many venues the participants were recruited from? Some detail is mentioned in the Results section but it is not very clear, e.g. a clinic waiting room (primary care clinic waiting room?), bars (what kind of bars? How many?). For example, how many transport hubs? How many trading centers? Additionally, what are the “other sites”?

Results: What is the “Figure: Pilot trial CONSORT diagram”? (p.9)

Results: It would be helpful if Table 1 was reformatted as it is currently challenging to read when the findings take the space of two rows.

Discussion: “individuals’ ability to make deposits may be limited by poverty and other circumstances.” Please provide an example of “other circumstances”. Also, are there any ethical issues to administering commitment contracts in low-and-middle income countries?

Discussion: “This approach also addressed a limitation observed in commitment contract programs in high-and middle-income countries…” Please expand on how this approach addressed this limitation?

Discussion: “was not powered to detect differences in retesting uptake…” Differences by study condition? Differences by participant characteristics? Please elaborate.

Discussion: “In future studies with more than one retesting opportunity, we suspect retesting might decline over time when incentives are not offered.” Please provide a reference here. Is this based on prior research?

Discussion: “Our pilot trial was designed to test acceptability and feasibility…” Is this paragraph a study limitations paragraph? If yes, please state more directly. For example, “These observations, along with the high prevalence of HIV among adults screened (15%...)…” It is not clear if this was stated as a study limitation or strength. Please clarify. Also, the study authors could mention if there were any study biases and how the study authors tried to respond or account for these.

Conclusion: “As reported rates of “ever” having tested…” As the sentence is currently structured, it seems like the study authors are stating that retesting is of “greater priority” than first-time testing. Is this what the study authors meant? Or that “retesting” is also of importance? Please clarify.

6. PLOS authors have the option to publish the peer review history of their article (what does this mean?). If published, this will include your full peer review and any attached files.

Reviewer #1: No

Reviewer #2: No

---

## [Author Response · Author response to Decision Letter 0]

30 Jan 2020

Response to Reviewers & Editor

Title: A pilot randomized trial of incentive strategies to promote HIV retesting

Manuscript: PONE-D-19-27665

Comments from the Editor:

Response: We have ensured that our manuscript meets PLOS ONE’s style requirements, including those for file naming.

2. Please provide additional details regarding participant consent. In the ethics statement in the Methods and online submission information, please ensure that you have specified whether consent was suitably informed.

Response: We now provide additional detail regarding participant consent in the Methods (see lines 169-170). All participants provided written informed consent in their preferred language (English or Runyankole, the local language in the region).

Response: Noted – thank you. We will provide the study dataset (Supporting Information 1: S1, noted on line 307) should our manuscript be accepted for publication.

4. Please include captions for your Supporting Information files at the end of your manuscript, and update any in-text citations to match accordingly. Please see our Supporting Information guidelines for more information:http://journals.plos.org/plosone/s/supporting-information

Response: We now provide captions for our Supporting Information files at the end of our manuscript (see lines 306-309).

 

Response to Reviewers’ Comments

Reviewer #1

1) Interesting study to incentivize retesting of negative individuals vs. incentivizing staying HIV negative. Considering the WHO 2015 Consolidated guidelines on HIV testing services and the upcoming update being released in December 2019 recommends that for most people who test HIV negative additional retesting to rule out being in the window period is not necessary (after 3 months). In addition, the WHO guidelines recommends retesting of individuals with ongoing high risk of HIV infection, not all key populations require retesting every 3 months. It would be helpful for the authors to address the above in the introduction and discussion.

Response: We now address, in both the Introduction and Discussion, retesting at 3-month intervals and annually (see Introduction, lines 61-66, and Discussion lines 284-292). We reference the WHO Consolidated HIV Testing Guidelines, and now include reference to the November 2019 update as well, which – though the update recommends annual retesting for all sexually active individuals in high HIV burden settings (such as Southwestern Uganda, where our pilot took place) – also notes that frequent retesting for HIV (i.e. every 3-6 months), “may be warranted based on individual risks and as part of broader HIV prevention interventions.” Uganda Ministry of Health guidelines at the time of this pilot (and currently) recommended retesting for HIV every 3 months in key populations – distinct from WHO guidelines, which we now address in the Introduction (see lines 61-66). In this pilot study, we worked with local health officials and community representatives in a generalized epidemic setting (Southwestern Uganda) to identify specific venues frequented by adults that the health officials/community leaders considered to be high-risk for HIV in their community. The HIV positivity of adults tested during recruitment (15%) is more than double that of the population HIV prevalence in the region (7%), suggesting that the input of local health officials/community leaders was accurately identifying high-risk venues. 

2) The methods of offering incentives is good; however, please address why incentivizing retesting of negative individuals vs. behavior change to remain negative.

Response: The reviewer correctly notes that various behaviors can be incentivized to prevent HIV, including a variety of prevention behaviors such as voluntary medical male circumcision, condom use, HIV retesting or other behaviors. However, many behaviors (condom use, changes in sexual behavior) cannot ethically or practically be observed or verified for the use of conditional financial incentives or deposit contracts. We therefore focused our efforts on a behavior we could verify (HIV retesting) and that can also act as a platform for ongoing counseling and linkage to other emerging combination prevention services over time.

3) Also address why incentivizing retesting of HIV negative individuals in a resource limited setting that is focusing more on targeted HIV testing services (e.g. partner notification services, risk screening tools in outpatient settings) with linkages to treatment and/or prevention services.

Response: Retesting of high-risk HIV negative individuals is a form of targeted HIV testing services, as reflected in Ugandan Ministry of Health guidelines for every 3-6 month retesting for key populations, such as the adults we recruited into our study. At the time of our pilot, pre-exposure prophylaxis (PrEP) was not available in government clinics; however, as PrEP becomes increasingly available, retesting at regular intervals (typically every 3 months in PrEP programs) will have even greater relevance among adults accessing PrEP services.

4) Please address how incentivizing retesting of the study participants is better than current practice in clinics or other HIV testing locations that offer HIV testing services? Please include references that show that there is a need to incentivize retesting of individuals who are HIV negative. There is a lot of peer reviewed literature that shows most HIV negative individuals (with or without ongoing risk) do come back for retesting.

Response: The primary objective of our pilot trial was not to directly compare the effectiveness of incentivizing HIV retesting vs. counseling participants to retest (current practice), and as such we cannot directly address how incentivizing retesting is better than current practice with our study data. However, regarding the need for interventions to promote retesting (the crux of the Reviewer’s insightful comment), peer-reviewed literature on retesting is fairly limited – but the several studies published on this topic have documented low rates of retesting at a frequency of at least annually (if not more frequently, such as every 3-6 months, as recommended for high-risk individuals: see response to first Reviewer #1 comment). For example, in a 2011 study that examined mobile testing outreach in South Africa, although 71% of adults tested reported any prior HIV test, only 37% reported testing within the prior 12 months (reference: Kranzer et al, High Prevalence of Self-Reported Undiagnosed HIV despite High Coverage of HIV Testing: A Cross-Sectional Population Based Sero-Survey in South Africa. PLoS One, 2011). Similarly, in a 2013 study from Tanzania evaluating individuals retesting for HIV over three time periods from 2003-10, only 11% of adults participated in retesting (reference: Cawley, C. et al. Low rates of repeat HIV testing despite increased availability of antiretroviral therapy in rural Tanzania: findings from 2003–2010. PloS ONE 8, e62212, 2013), and in a 2013 study, Regan et al found only 26% of adults reporting retesting in two South African hospitals (reference: Regan et al, Factors Associated with Self-Reported Repeat HIV Testing after a Negative Result in Durban, South Africa, PLoS One, 2013). Based on our prior work in Southwestern Uganda that sought to recruit all adult men across a community for HIV testing, though any prior HIV testing was common (78% of men), less than half reported testing within the past year (Chamie, et al, AIDS, 2018). These published data highlight low rates of HIV retesting, falling well below WHO recommendations of annual retesting for all adults, let alone more frequent testing of high-risk adults. Lastly, in a recent systematic review of interventions to promote retesting, the authors noted that, “evidence-based interventions that promote frequent and repeat testing more than once a year among individuals at high risk for HIV remain sparse,” (reference: Paschen-Wolff, et al, A Systematic Review of Interventions that Promote Frequent HIV Testing, AIDS and Behavior, 2019), emphasizing the need to better study interventions to promote retesting.

We now review this literature in greater detail in the revised Introduction section (see lines 67-79, and 80-92).

 

Reviewer #2: 

General Comments:

Background: The study authors outline how the use of financial incentives in the role of HIV re-testing have yet to be studied. While this might be the case, I think the authors could make a stronger case for why the use of financial incentives for retesting would expect to be any different than the use of financial incentives for HIV testing in general? For example, do the authors expect a different outcome than what research has shown with financial incentives for HIV testing? Please clarify.

Response: We have now clarified in greater detail why retesting for HIV after a negative test is likely to be a distinct behavior from incentivizing HIV testing in general (see Introduction, lines 80-92). Motivation to retest for HIV after a negative test may be lower than testing for HIV in general, given studies that have shown perceptions that retesting is unnecessary (e.g. Orne-Gliemann, J., et al. (2016). "Community perceptions of repeat HIV-testing: experiences of the ANRS 12249 Treatment as Prevention trial in rural South Africa." AIDS care 28 Suppl 3: 14-23).

Background: This manuscript would benefit from some additional references, particularly in the Background section to provide a stronger case for how the use of financial incentives for HIV retesting would be beneficial for the field. For example, what is the prevalence and/or incidence of HIV infection in Uganda? How often do people get retested in Uganda? If the study authors provided some brief background information about HIV and HIV retesting uptake in Uganda this might also increase clarity for the international reader as to why innovative interventions are needed in this country/region.

Response: We now provide additional references and information in our Introduction section to offer more background information about the need for interventions to promote retesting for HIV, and about HIV and retesting uptake among high-risk adults available from published studies in sub-Saharan Africa, including Uganda. (See Introduction, lines 67-79, and 80-92). 

Discussion: Please provide some detail as to how the study findings are comparable to other studies that use financial incentives in low-and-middle income countries in the HIV field.

Response: We now include greater detail on how our study findings are comparable to other studies that have used incentives for HIV testing in low- and middle-income settings, including reference to a systematic review of incentives for HIV testing (see Discussion, lines 251-270).

Discussion: Please provide a brief statement how the findings from this study could inform future research in the field, and also, remaining knowledge gaps and areas for future research.

Response: We now provide a brief statement about how the findings from this study could inform future research in the field and have provide knowledge gaps remain as a basis for areas of future research (see Discussion, lines 265-270).

Specific Comments:

Abstract: In the Results section, the authors have listed number/percentage to display one set of findings but then only percentages for the remaining findings. Please use number/percentage to be consistent.

Response: We now provide number/percentages throughout the Results section of the Abstract for consistency.

Introduction: “Economic incentives have been shown to promote a number of behaviours…” This paragraph is not entirely clear. For example, is ‘counseling alone’ the standard of care? 

Response: We have revised this paragraph (see Introduction, lines 80-92) for greater clarity, including the standard of care, which is counseling alone in this setting.

Please rephrase this sentence: “the costs of a given behavior are immediate but the gains may not be realized until later…” (could the authors please provide an example?). Also, with this sentence: “insights from behavioural economic theory suggest that individuals display loss aversion…” Perhaps the authors provide an example to increase clarity of “loss aversion”?

Response: We have rephrased these sentences highlighted by the reviewer, providing examples for the phrase discussing “costs of a given behavior”, as well as an example to clarify loss aversion (see Introduction, lines 80-84, and lines 93-99).

Introduction: “A standard approach to implementing…” Please provide some references here to increase clarity.

Response: We now provide references for this section (Introduction, lines 93-95).

Introduction: “Individuals may also be aware of a tendency towards present-biased…” Please modify this sentence to increase clarity.

Response: We have revised this sentence in an effort to increase clarity (lines 99-110).

Introduction: What is “Leverage loss aversion”? Please clarify.

Response: We have revised this sentence in an effort to increase clarity (lines 95-99).

Methods: How long did the authors give the participants to be re-tested? Did participants have to come back at exactly 3 months post study enrollment for re-testing? Please add some study details to increase clarity.

Response: As we note in the Methods (please see Methods, lines 155-157) participants did not have to return for retesting at exactly 3 months post-enrollment but were instead provided an interval of 2-3 months to return to the clinic for retesting.

Methods: “considered at increased risk of HIV, due to drinking, sex work” What does “due to drinking” mean in this case? Please remove or modify to increase clarity.

Response: We now have modified this phrase for further clarity (see Methods, lines 126-128). The phrase now states, “due to drinking at bars associated with transactional sex.”

Methods: What was the questionnaire that participants were asked to complete? Did the questionnaire only enquire about participant demographics or also measure ‘feasibility’ and ‘acceptability’? How many questions were in the questionnaire? Who developed the questionnaire? Who administered the questionnaire? Was it self-administered?

Response: The questionnaire was developed by the investigators, administered by study staff, and sought to gather information on both demographics, socioeconomic status, HIV risk behavior, prior HIV testing behavior and testing preferences. There were 62 questions in the questionnaire. These details are now provided in the Methods, (see lines 143-145). 

Methods: Please specifically define ‘feasibility’ and ‘acceptability’ and provide a reference, if possible. Also, please define how these outcomes are measured. In the Discussion section, it states, “were feasible and had high acceptability”, however, it is not clear how these outcomes were measured (and what could be defined as ‘high acceptability’).

Response: We now define acceptability in our methods (see lines 160-162. We pre-specified our definition of acceptability and feasibility in our study protocol (included with our PLoS One submission). Feasibility and how we measured these outcomes are clearly defined in our Methods as well (see lines 162-163). As this was a pilot trial, we did not pre-specify the definition of “high” acceptability, but we now clarify in our Discussion (line 221) that when we indicate “high acceptability”, this refers to the finding that 95% of eligible participants agreed to proceed with enrollment and continued with trial participation after randomization.

Methods: Please define “standard, gain framed cash incentive” to increase clarity. Similarly, what is a “gain framed” incentive?

Response: A “gain-framed” incentive is a simple financial reward (or bonus) provided for doing a given, desired behavior. We now clarify this phrase (see line 150).

Methods: Where did participants need to go to receive retesting? At one clinic site? At several clinic sites?

Response: Participants needed to return to the same, single clinic site to receive retesting. We now clarify this point in the Methods (see lines 159-160).

Methods: What is standard of care in this case? i.e. “no incentive (control)”

Response: The standard of care is counseling to promote retesting without the use of an incentive (i.e. what we provided in our control arm: no incentive, but counseling) – and this standard is now clarified in our Introduction (see lines 90-91).

Methods: “we determined sample size…in feasibility of commitment contracts in approximately 70 adults rather than using power calculations…” Please consider modifying this section to increase clarity. It is not clear how the study authors determined a sample size of 70 to measure feasibility. Further, was the sample size determined to only measure feasibility or also acceptability? At this point, the study authors could state the minimum number of participants needed in each condition to be powered to detect statistical significant differences between conditions. The study authors could also emphasise in the Results section that the study was not powered to detect statistical significance between conditions, and in the Discussion section, mention that this is an area for future research (would be powered in future trial).

Response: We did not use power calculations to determine a sample size, and we feel that presenting the sample sizes needed to detect statistical differences between arms (which is a total sample size of 525 participants in our subsequent 3-arm trial to promote retesting) would introduce more confusion than clarity. We now emphasize in our Results section that the study was not powered to detect statistically significant differences in retesting between arms (see lines 214-215) and mention that this is an area for future research in the Discussion section (see lines 224-225).

Methods: Could the study authors please provide greater details as to the venues in which the participants were recruited from? Is there any indication of how many venues the participants were recruited from? Some detail is mentioned in the Results section but it is not very clear, e.g. a clinic waiting room (primary care clinic waiting room?), bars (what kind of bars? How many?). For example, how many transport hubs? How many trading centers? Additionally, what are the “other sites”?

Response: We now include greater detail on the venues from which participants were recruited in the Methods section (see page 7, lines 127-132). We also now include greater detail on the venues in our Results section, including clarification as to what the “other sites” were and the number of venues (see page 10, lines 179-182).

Results: What is the “Figure: Pilot trial CONSORT diagram”? (p.9)

Response: This Figure was submitted with the manuscript, and is a CONSORT diagram, now referenced, that is recommended to use to provide detail on recruitment, enrollment and retention of participants in pilot trials. We have elaborated on this in the description of the Figure (see lines 201-202).

Results: It would be helpful if Table 1 was reformatted as it is currently challenging to read when the findings take the space of two rows.

Response: The formatting is based on PLoS One requirements to have double spacing of tables and is not how the table will appear if accepted for publication by PLoS One. Ultimately, if the manuscript is accepted, we will request that the table have findings on a single row.

Discussion: “individuals’ ability to make deposits may be limited by poverty and other circumstances.” Please provide an example of “other circumstances”. Also, are there any ethical issues to administering commitment contracts in low-and-middle income countries?

Response: We have removed the phrase, “other circumstances,” as we feel poverty is the primary driver of one’s ability to make a deposit (see line 232). As commitment contracts are voluntary, and suggested deposits are considered low, we do not believe there are any ethical issues to administering commitment contracts.

Discussion: “This approach also addressed a limitation observed in commitment contract programs in high-and middle-income countries…” Please expand on how this approach addressed this limitation?

Response: We now expand on how this approach addressed this limitation (see Discussion, lines 236-238).

Discussion: “was not powered to detect differences in retesting uptake…” Differences by study condition? Differences by participant characteristics? Please elaborate.

Response: We now clarify this phrase, as it was used to indicate detection of differences by study condition (see lines 273).

Discussion: “In future studies with more than one retesting opportunity, we suspect retesting might decline over time when incentives are not offered.” Please provide a reference here. Is this based on prior research?

Response: This suspicion that retesting might decline over time is based on the observation that with healthy behaviors that require repeated/continued engagement (such as smoking cessation or exercise), the withdrawal of incentives can lead to a decline in healthy behavior from the levels of behavior achieved with an incentive. We also now include an example and a reference for this tendency observed in prior research in the manuscript (see line 279).

Discussion: “Our pilot trial was designed to test acceptability and feasibility…” Is this paragraph a study limitations paragraph? If yes, please state more directly. For example, “These observations, along with the high prevalence of HIV among adults screened (15%...)…” It is not clear if this was stated as a study limitation or strength. Please clarify. Also, the study authors could mention if there were any study biases and how the study authors tried to respond or account for these.

Response: We now directly state that this paragraph is a study limitations paragraph, and enumerate our limitations (“First”, “Second”, “Third”, etc.), to distinguish them from the strengths we highlight in response to these limitations (see lines 271-296).

Conclusion: “As reported rates of “ever” having tested…” As the sentence is currently structured, it seems like the study authors are stating that retesting is of “greater priority” than first-time testing. Is this what the study authors meant? Or that “retesting” is also of importance? Please clarify.

Response: We are not stating that retesting is of greater priority than first-time testing in general, but rather making the point that as rates of “ever testing” reach very high levels, the relative importance of retesting increases. We now clarify this point (see lines 299-300).

---

## [Decision Letter · Decision Letter 1]

19 Feb 2020

PONE-D-19-27665R1

A pilot randomized trial of incentive strategies to promote HIV retesting

PLOS ONE

Dear Dr. Chamie,

Thank you for submitting your manuscript to PLOS ONE. After careful consideration, we feel that it has merit but does not fully meet PLOS ONE’s publication criteria as it currently stands. Therefore, we invite you to submit a revised version of the manuscript that addresses the points raised during the review process.

We really appreciate the efforts made by the authors, but as you will see, the Reviewer 2 still has concerns that require response.

We would appreciate receiving your revised manuscript by March 30. To enhance the reproducibility of your results, we recommend that if applicable you deposit your laboratory protocols in protocols.io, where a protocol can be assigned its own identifier (DOI) such that it can be cited independently in the future. For instructions see: http://journals.plos.org/plosone/s/submission-guidelines#loc-laboratory-protocols

We look forward to receiving your revised manuscript.

Kind regards,

Giuseppe Vittorio De Socio, MD, PhD

Academic Editor

PLOS ONE

Reviewers' comments:

Reviewer's Responses to Questions

**Comments to the Author**

1. If the authors have adequately addressed your comments raised in a previous round of review and you feel that this manuscript is now acceptable for publication, you may indicate that here to bypass the “Comments to the Author” section, enter your conflict of interest statement in the “Confidential to Editor” section, and submit your "Accept" recommendation.

Reviewer #1: All comments have been addressed

Reviewer #2: All comments have been addressed

2. Is the manuscript technically sound, and do the data support the conclusions?

Reviewer #1: Partly

Reviewer #2: Partly

3. Has the statistical analysis been performed appropriately and rigorously? 

Reviewer #1: Yes

Reviewer #2: Yes

4. Have the authors made all data underlying the findings in their manuscript fully available?

Reviewer #1: Yes

Reviewer #2: Yes

5. Is the manuscript presented in an intelligible fashion and written in standard English?

Reviewer #1: Yes

Reviewer #2: Yes

6. Review Comments to the Author

Reviewer #1: It is technically sound and all comments were addressed. However, incentivizing retesting of negative individuals vs. prevention to remain negative in resource limited settings is not sustainable and somewhat conflicts the updated 2019 consolidated guidelines recommendations on targeted testing in changing epidemics.

Reviewer #2: The study authors have made some substantial changes to this manuscript, which has helped to increase clarity. I have provided a few more suggested modifications for your consideration:

Title: The title is a bit too concise. Please consider adding the country/location of the study and/or the study population group. For example, “to promote HIV restesting among …. In Uganda”

Abstract: “Retesting for HIV is critical to identifying newly-infected persons and reinforcing prevention…” The ‘reinforcing’ term is unclear. Do the authors mean reinforcing prevention efforts? Prevention messaging? Please modify.

Abstract: “Incentives can increase one-time testing…” Based on? Perhaps better to put ‘may increase’?

Entire Manuscript: As much as possible, it is better to use ‘HIV retesting’ instead of ‘retesting’ as a firm reminder to the reader that the authors are referring to retesting of HIV. For example, Pg. 9 “the deposits would be lost if they did not return for HIV retesting …”. Also, Pg.14, “clear understanding of the benefits of HIV retesting …”

Abstract: “to promote retesting three months later…” Do the authors mean, three months post-study enrolment? Please clarify.

Abstract: “Contracts sought to promote retesting by leveraging …” The study authors explain this sentence in the manuscript, so fine to leave there, but it is a bit confusing in the Abstract. Could remove.

Abstract: “(proportion making deposits)” – Perhaps, ‘proportion of participants making deposits’?

Abstract: “Overall, 93 (76%) participants retested” – Please consider adding “were retested for HIV” – Again, making clear to the reader that we are speaking to retesting of HIV.

Introduction: The first section of the Introduction is verbose and hard to understand. Please modify. For example, what are combination prevention approaches?

Introduction: “Retesting of high-risk, HIV-uninfected persons is critical to…” Please add a reference after this point.

Introduction: “Retesting also offer an opportunity to …” Please add a reference after this point.

Introduction: “may be perceived as low, particularly if access” Access to what? A clinic? Please clarify.

Introduction: “incentives can be made more powerful”. I am not sure that ‘powerful’ is the right word here. Do the authors mean ‘effective’ or ‘impactful’?

Introduction: “Recent studies have indicated that …” Please add the references to the ‘recent studies’ here.

Introduction: “Very few incentive-based approaches…” Please add references here to support this statement.

Introduction: “We assessed the feasibility and acceptability …” This statement could be made stronger by stating, “The aim of this study is to…”

Methods: “due to drinking at bars associated with transactional sex…” This sentence is unclear. Do the authors mean that the “adults considered at increased risk of HIV due to attendance at … and behaviours in sex work …?” Please modify.

Methods: “rapid HIV antibody testing”. Could the study authors please list the type of test that was used (e.g. brand name, etc.).

Results: “HIV positivity varied by recruitment site …” Was this variance expected? If no, please list this more directly in the Limitations section. For example, following the sentence, “Possibly suggesting that we did not reach a high-risk population group through our selection of recruitment study sites …”

Results: Please list as ‘Figure 1’ and ‘Table 1’.

Results: Figure 1 is quite helpful and should be included as part of the manuscript instead of supplementary material, if possible (i.e. it was not embedded in the manuscript like the Table).

Discussion: “the optimal balance between deposit amounts that are acceptable versus of sufficient value to generate …” I know what the authors are trying to say, but I think it is the ‘versus’ that makes the statement a bit awkward. Could change to ‘that are acceptable and of sufficient value’.

Discussion: “other HIV-related behaviours and outcomes, such as voluntary medical male …” If the study authors keep the examples of VMMC and HIV viral suppression in this specific sentence, then references should be used. The authors could also remove these specific examples in this sentence, given that the examples are provided in the following sentence with references. It keeps the overall paragraph less repetitive. For example, “other HIV-related behaviours and outcomes. The effectiveness of these strategies when studied …Examples of effective strategies have included …”

Discussion: “decreasing engagement in risky sex among men…” This sentence is a bit unclear. Is this referring to men who have sex with men? Or more that men are engaging in high-risk sexual activities? Please clarify.

Discussion: “whether commitment contract participants…” This is a bit of lengthy adjective. Perhaps change to “whether participants who agree to commitment contracts…”

Discussion: “Fourth, no seroconversions were observed…” Again, re-emphasise HIV. “Fourth, there were no HIV seroconversions observed in this study …”

Discussion: “However, participants reported high-risk sexual behaviour…” The sexual behaviours of the participants were presented briefly in Table 1. Are there other components of the questionnaire that can be mentioned in paragraph 1 or 2 of the Results section or placed in Table 1? (but not to be mentioned in both places). The authors used a 62-item questionnaire, so there could be more Results presented from the questionnaire in either Table 1 or written in the Results section. The authors could also provide a copy of the 62-item questionnaire in the Supplementary information section.

It might be clearer if the study authors use ‘Table 1’ to highlight baseline characteristics of study participants (study demographics, sexual activity behaviours, etc.) and ‘Table 2’ to show study outcomes (e.g. ‘HIV retesting 2-3 months following randomisation’ ‘Deposit made to commitment contract’) because the title of Table 1 currently is baseline characteristics, and HIV retesting, for example, is not a baseline characteristic. There are also empty spaces in some parts of Table 1 and then ‘N/A’ in other sections, would be more helpful to be consistent.

7. PLOS authors have the option to publish the peer review history of their article (what does this mean?). If published, this will include your full peer review and any attached files.

Reviewer #1: No

Reviewer #2: No

---

## [Author Response · Author response to Decision Letter 1]

24 Mar 2020

Response to Reviewers & Editor

Title: A pilot randomized trial of incentive strategies to promote HIV retesting

Manuscript: PONE-D-19-27665

Response to Reviewers’ Comments

Reviewer #1

General Comments:

It is technically sound and all comments were addressed. However, incentivizing retesting of negative individuals vs. prevention to remain negative in resource limited settings is not sustainable and somewhat conflicts the updated 2019 consolidated guidelines recommendations on targeted testing in changing epidemics.

Response: We are pleased that Reviewer 1 finds all of the comments addressed and the manuscript technically sound. The use of incentives are endorsed by the WHO 2019 Consolidated guidelines, and could be used to increase retesting that is targeted to very high-risk people who test negative for HIV, but based on risk behavior, merit retesting 3-6 months in the future – a recommendation in line with WHO guidelines. Furthermore, deposit contracts are potentially cost saving compared to standard incentives, as they leverage voluntary deposits by adults who wish to participate in them. However, our manuscript is not designed to evaluate or test the sustainability of this approach, but rather to understand the feasibility and acceptability of incentive strategies to promote HIV retesting. We agree that further research is needed to address the question of cost-effectiveness and sustainability.

Reviewer #2: 

General Comments:

The study authors have made some substantial changes to this manuscript, which has helped to increase clarity. I have provided a few more suggested modifications for your consideration:

Specific Comments:

Title: The title is a bit too concise. Please consider adding the country/location of the study and/or the study population group. For example, “to promote HIV restesting among …. In Uganda” 

Response: We have revised the title to include the study location as rural Uganda. 

Abstract: “Retesting for HIV is critical to identifying newly-infected persons and reinforcing prevention…” The ‘reinforcing’ term is unclear. Do the authors mean reinforcing prevention efforts? Prevention messaging? Please modify. Page 4 of 6 

Response: We have revised this sentence to clarify that HIV retesting can reinforce prevention efforts among at-risk adults. 

Abstract: “Incentives can increase one-time testing…” Based on? Perhaps better to put ‘may increase’? 

Response: This statement in the abstract is based on multiple studies, including trials and a systematic review, that have demonstrated that incentives can increase one-time HIV testing. We provide these references in our Introduction section (page 6, lines 81-85). 

Entire Manuscript: As much as possible, it is better to use ‘HIV retesting’ instead of ‘retesting’ as a firm reminder to the reader that the authors are referring to retesting of HIV. For example, Pg. 9 “the deposits would be lost if they did not return for HIV retesting …”. Also, Pg.14, “clear understanding of the benefits of HIV retesting …” 

Response: We have revised the manuscript to emphasize that our aim and outcome was increase in HIV retesting. These clarifications include the reviewer’s specific suggestions to add “HIV” to the term “retesting” on pages 9 and 14, as well as throughout the manuscript. 

Abstract: “to promote retesting three months later…” Do the authors mean, three months post-study enrolment? Please clarify. 

Response: We have revised this sentence to clarify that our outcome was HIV retesting at three months after study enrollment.

Abstract: “Contracts sought to promote retesting by leveraging …” The study authors explain this sentence in the manuscript, so fine to leave there, but it is a bit confusing in the Abstract. Could remove. 

Response: We recognize that this term is not fully defined in the abstract, but it is a well-known behavioral economic term that is well-defined in our manuscript, and key to how our deposit intervention works, so we have chosen to leave it in the manuscript.

Abstract: “(proportion making deposits)” – Perhaps, ‘proportion of participants making deposits’? 

Response: We have revised this sentence to clarify that feasibility was assessed as the proportion of participants who chose to make a deposit.

Abstract: “Overall, 93 (76%) participants retested” – Please consider adding “were retested for HIV” – Again, making clear to the reader that we are speaking to retesting of HIV. 

Response: We have revised this sentence to clarify that our outcome was HIV retesting uptake. 

Introduction: The first section of the Introduction is verbose and hard to understand. Please modify. For example, what are combination prevention approaches? 

Response: We now provide some examples of combination HIV prevention approaches to provide greater clarity to the first paragraph of the Introduction.

Introduction: “Retesting of high-risk, HIV-uninfected persons is critical to…” Please add a reference after this point. 

Response: We now provide references for this sentence.

Introduction: “Retesting also offer an opportunity to …” Please add a reference after this point. 

Response: We now provide a reference for this point.

Introduction: “may be perceived as low, particularly if access” Access to what? A clinic? Please clarify. 

Response: This sentence has been revised to specify that access to testing and transportation to clinics may be costly for some individuals. 

Introduction: “incentives can be made more powerful”. I am not sure that ‘powerful’ is the right word here. Do the authors mean ‘effective’ or ‘impactful’? 

Response: This sentence has been revised to clarify that the use of loss aversion principles can make incentives more effective. 

Introduction: “Recent studies have indicated that …” Please add the references to the ‘recent studies’ here.

Response: We now provide references for this sentence.

Introduction: “Very few incentive-based approaches…” Please add references here to support this statement. 

Response: This phrase is based on our own extensive review of the literature, and does not have a specific reference, as currently there is no systematic review of incentive-based approaches that leverage loss aversion. We have revised this sentence to clarify that, “to our knowledge, no incentive-based approaches in sub-Saharan Africa have attempted to leverage loss aversion for behavior change through commitment contracts.” 

Introduction: “We assessed the feasibility and acceptability …” This statement could be made stronger by stating, “The aim of this study is to…” 

Response: This sentence has been revised, based on the reviewer’s suggestion to strengthen the message and highlight the study aim. 

Methods: “due to drinking at bars associated with transactional sex…” This sentence is unclear. Do the authors mean that the “adults considered at increased risk of HIV due to attendance at … and behaviours in sex work …?” Please modify. 

Response: This sentence has been revised to clarify that individuals who attended bars and sites of sex work were considered to be at increased risk of HIV-infection and were specifically recruited for this study. 

Methods: “rapid HIV antibody testing”. Could the study authors please list the type of test that was used (e.g. brand name, etc.). 

Response: The study performed HIV antibody testing using the Alere DetermineTM HIV-1/2 Ag/Ab Combo, followed by the Chembio HIV 1/2 STAT-PAK® and SD Bioline HIV-1/2 assays for confirmation of Determine-positive results. This information has been added to the Methods section.

Results: “HIV positivity varied by recruitment site …” Was this variance expected? If no, please list this more directly in the Limitations section. For example, following the sentence, “Possibly suggesting that we did not reach a high-risk population group through our selection of recruitment study sites …” 

Response: This variance was expected. 

Results: Please list as ‘Figure 1’ and ‘Table 1’. 

Response: The figure and table labels have been revised to read “Figure 1” and “Table 1” as suggested. 

Results: Figure 1 is quite helpful and should be included as part of the manuscript instead of supplementary material, if possible (i.e. it was not embedded in the manuscript like the Table). 

Response: We are pleased that Reviewer 2 finds Figure 1 helpful. Figure 1 will be included as part of the manuscript (it is not supplementary material). However, PlosOne requires that figures be uploaded as separate files and not be embedded in the manuscript.

Discussion: “the optimal balance between deposit amounts that are acceptable versus of sufficient value to generate …” I know what the authors are trying to say, but I think it is the ‘versus’ that makes the statement a bit awkward. Could change to ‘that are acceptable and of sufficient value’. Page 5 of 6 

Response: This sentence has been revised to clarify that future studies may need to determine the optimal balance between deposit contract amounts that are acceptable and of sufficient value to generate loss aversion. 

Discussion: “other HIV-related behaviours and outcomes, such as voluntary medical male …” If the study authors keep the examples of VMMC and HIV viral suppression in this specific sentence, then references should be used. The authors could also remove these specific examples in this sentence, given that the examples are provided in the following sentence with references. It keeps the overall paragraph less repetitive. For example, “other HIV-related behaviours and outcomes. The effectiveness of these strategies when studied …Examples of effective strategies have included …” 

Response: Voluntary medical male circumcision and HIV viral suppression have been removed from the sentence indicated in this comment from the reviewer. These specific examples have been retained later in the same paragraph with references cited. 

Discussion: “decreasing engagement in risky sex among men…” This sentence is a bit unclear. Is this referring to men who have sex with men? Or more that men are engaging in high-risk sexual activities? Please clarify. 

Response: This sentence has been clarified to indicate that the study in question assessed high-risk sexual activity among men in Malawi. 

Discussion: “whether commitment contract participants…” This is a bit of lengthy adjective. Perhaps change to “whether participants who agree to commitment contracts…” 

Response: This sentence has been revised as suggested by the reviewer.

Discussion: “Fourth, no seroconversions were observed…” Again, re-emphasise HIV. “Fourth, there were no HIV seroconversions observed in this study …” 

Response: This sentence has been revised to emphasize that the outcome being observed was HIV seroconversions.

Discussion: “However, participants reported high-risk sexual behaviour…” The sexual behaviours of the participants were presented briefly in Table 1. Are there other components of the questionnaire that can be mentioned in paragraph 1 or 2 of the Results section or placed in Table 1? (but not to be mentioned in both places). The authors used a 62-item questionnaire, so there could be more Results presented from the questionnaire in either Table 1 or written in the Results section. The authors could also provide a copy of the 62-item questionnaire in the Supplementary information section. 

Response: The questions regarding sexual risk that we asked in our 62-item questionnaire were brief and are all included in Table 1. We now provide a copy of the 62-item questionnaire for inclusion in the Supplementary materials.

It might be clearer if the study authors use ‘Table 1’ to highlight baseline characteristics of study participants (study demographics, sexual activity behaviours, etc.) and ‘Table 2’ to show study outcomes (e.g. ‘HIV retesting 2-3 months following randomisation’ ‘Deposit made to commitment contract’) because the title of Table 1 currently is baseline characteristics, and HIV retesting, for example, is not a baseline characteristic. There are also empty spaces in some parts of Table 1 and then ‘N/A’ in other sections, would be more helpful to be consistent.

Response: We have opted to present a single table, as the retesting outcomes are brief (3 rows), and the title of Table 1 is “Baseline characteristics and HIV retesting uptake,” indicating that both outcomes are presented in this single table. We have added “N/A” to the empty spaces and thank the Reviewer for this suggestion.

---

## [Decision Letter · Decision Letter 2]

21 Apr 2020

PONE-D-19-27665R2

A pilot randomized trial of incentive strategies to promote HIV retesting in rural Uganda

PLOS ONE

Dear Dr. Chamie,

Thank you for submitting your manuscript to PLOS ONE. After careful consideration, we feel that it has merit but does not fully meet PLOS ONE’s publication criteria as it currently stands. Therefore, we invite you to submit a revised version of the manuscript that addresses the points raised during the review process.

We would appreciate receiving your revised manuscript by the next 4 weeks. To enhance the reproducibility of your results, we recommend that if applicable you deposit your laboratory protocols in protocols.io, where a protocol can be assigned its own identifier (DOI) such that it can be cited independently in the future. For instructions see: http://journals.plos.org/plosone/s/submission-guidelines#loc-laboratory-protocols

We look forward to receiving your revised manuscript.

Kind regards,

Giuseppe Vittorio De Socio, MD, PhD

Academic Editor

PLOS ONE

Reviewers' comments:

Reviewer's Responses to Questions

**Comments to the Author**

1. If the authors have adequately addressed your comments raised in a previous round of review and you feel that this manuscript is now acceptable for publication, you may indicate that here to bypass the “Comments to the Author” section, enter your conflict of interest statement in the “Confidential to Editor” section, and submit your "Accept" recommendation.

Reviewer #3: (No Response)

2. Is the manuscript technically sound, and do the data support the conclusions?

Reviewer #3: Yes

3. Has the statistical analysis been performed appropriately and rigorously? 

Reviewer #3: Yes

4. Have the authors made all data underlying the findings in their manuscript fully available?

Reviewer #3: Yes

5. Is the manuscript presented in an intelligible fashion and written in standard English?

Reviewer #3: Yes

6. Review Comments to the Author

Reviewer #3: This is a very interesting study assessing the feasibility and acceptability of various incentive strategies, including commitment contracts that required a baseline deposit, to promote HIV retesting among at-risk adults in rural Uganda.

Some minor comments considered important.

Although the methods sections is well detailed, the authors have not explicitly included a statistical analysis section. For example descriptive statistics presented as means(SD), median(IQR) and counts(%ages), also the authors have presented proportions with 95%CI using exact methods to obtain binomial confidence intervals. All this should be stated in the analysis section after methods or within the methods section.

There were a lot of baseline questionnaires collected but only a selection of variables have been presented in Table 1, for an example education, occupation, marital status, socio-economic status, as the readers gain more understanding of the characteristics of individuals who are identified as high risk for HIV infections. These categories could be grouped into sensible groups ofcourse. These variables can added as an additional table in the manuscript.

Table 1: The variables “Daily wage (2017 US Dollars): mean”, please include SDs as well.

7. PLOS authors have the option to publish the peer review history of their article (what does this mean?). If published, this will include your full peer review and any attached files.

Reviewer #3: No

---

## [Author Response · Author response to Decision Letter 2]

6 May 2020

Response to Reviewers & Editor

Title: A pilot randomized trial of incentive strategies to promote HIV retesting

Manuscript: PONE-D-19-27665

Response to Reviewers’ Comments

Reviewer #3

General Comments:

This is a very interesting study assessing the feasibility and acceptability of various incentive strategies, including commitment contracts that required a baseline deposit, to promote HIV retesting among at-risk adults in rural Uganda. Some minor comments considered important.

Specific Comments:

Although the methods sections is well detailed, the authors have not explicitly included a statistical analysis section. For example descriptive statistics presented as means(SD), median(IQR) and counts(%ages), also the authors have presented proportions with 95%CI using exact methods to obtain binomial confidence intervals. All this should be stated in the analysis section after methods or within the methods section.

Response: We have revised the methods section to include a statistical analysis section (under a subheading within the Methods section), including additional detail about the statistical analyses performed for this study. 

There were a lot of baseline questionnaires collected but only a selection of variables have been presented in Table 1, for an example education, occupation, marital status, socio-economic status, as the readers gain more understanding of the characteristics of individuals who are identified as high risk for HIV infections. These categories could be grouped into sensible groups of course. These variables can added as an additional table in the manuscript.

Response: We now include baseline education and marital status in Table 1, and socio-economic status is included as mean daily wage. 

Table 1: The variables “Daily wage (2017 US Dollars): mean”, please include SDs as well.

Response: We now included standard deviations for these resulted variables.

---

## [Editor Report · Decision Letter 3]

11 May 2020

A pilot randomized trial of incentive strategies to promote HIV retesting in rural Uganda

PONE-D-19-27665R3

Dear Dr. Chamie,

We are pleased to inform you that your manuscript has been judged scientifically suitable for publication and will be formally accepted for publication once it complies with all outstanding technical requirements.

With kind regards,

Giuseppe Vittorio De Socio, MD, PhD

Academic Editor

PLOS ONE
---

## [Editor Report · Acceptance letter]

20 May 2020

PONE-D-19-27665R3 

A pilot randomized trial of incentive strategies to promote HIV retesting in rural Uganda 

Dear Dr. Chamie:

I am pleased to inform you that your manuscript has been deemed suitable for publication in PLOS ONE. Congratulations! Your manuscript is now with our production department. 

With kind regards,

on behalf of

Dr. Giuseppe Vittorio De Socio 

Academic Editor

PLOS ONE